# Open Innovation 4.0 as an Enhancer of Sustainable Innovation Ecosystems

**Joana Costa [1,2,\*] and João C.O. Matias [1,2]** 

[1] DEGEIT—Department of Economics, Management, Industrial Engineering and Tourism, University of Aveiro, Campus Universitário de Santiago, 3810-193 Aveiro, Portugal; jmatias@ua.pt

[2] GOVCOPP—Research Unit on Governance, Competitiveness and Public Policies, Campus Universitário de Santiago, 3810-193 Aveiro, Portugal

\* Correspondence: joanacosta@ua.pt

**Abstract:** Innovation matters. Business success increasingly depends upon sustainable innovation. Observing recent innovation best practices, the emergence of a new paradigm is traceable. Creating an innovative ecosystem has a multilayer effect: It contributes to regional digitalization, technological start-up emergence, open innovation promotion, and new policy enhancement retro-feeding the system. Public policy must create open innovation environments accordingly with the quintuple helix harmonizing the ecosystem to internalize emerging spillovers. The public sector should enhance the process, providing accurate legal framework, procurement of innovation, and shared risks in R&D. Opening the locks that confine the trunks of community, academic, industry, and government innovation will harness each dimension exploiting collective and collaborative potential of individuals towards a brighter sustainable future. In this sense, the aim of this study is to present how open innovation can enhance sustainable innovation ecosystems and boost the digital transition. For that, firstly, a diachronic perspective of the sustainable innovation ecosystem is traced, its connection to open innovation, and identification of the university linkages. Secondly, database exploration and econometric estimations are performed. Then, we will ascertain how far open innovation frameworks and in particular the knowledge flows unveiled by the university promote smart and responsible innovation cycles. Lastly, we will propose a policy package towards green governance, empowering the university in governance distributed ecosystem, embedded in the community, self-sustained with shared gains, and a meaningful sense of identity.

**Keywords:** open innovation; innovation ecosystems; sustainability; logit models

## 1. Introduction

Challenges faced worldwide are too large to tackle in isolation; the creation of new shared value through innovation is urgent. Convergent globalized features likewise digital transformation, universal interaction, and sustainability provided momentum to exponential growth in innovation led shared value. Approaching the new innovation paradigms requires integrated collaboration, co-creation, value sharing, hosting ecosystems, and fast adoption; joint research will speed up the process and raise the standards of the outcomes. Dissimilarly from previous industrial shifts, Industry 4.0 relies on pairing and amalgamation of subjects and expertise [1].

Creating shared value, sustainable growth, and development, relying on innovation, will overflow the economic and social sphere as under the collaborative paradigm firms will move from the maximization of short-term financial performance to long-term economic and social responsibility [2].

To present standards, assessing innovative performance based on patent count or R&D expenditures seems to be insufficient; co-creation emerging from knowledge network connections

seems to be more robust, based upon the interactive contact between universities providing cutting edge knowledge and competitive firms [3]. Therefore, the development of innovation demands a particular ecosystem in which they will emerge as a result of the collaboration and co-creation among different players. The ecosystem approach emphasizes the position and roles of local and public actors in developing innovative activities, and the public policy challenge is to provide the means and instruments to transform traditional environments in innovative milieu [4].

The purpose of this research is twofold: At first, identifying the relevant players in the promotion of sustainable innovation ecosystems as the underlying foundation for the digital transformation and followed by investigating if their role does hold for the different innovation types. The second is to further explore the importance of relating open innovation to the academia and the user community to establish self-sustained networks and propose a policy package to support these ecosystems along the digital transformation. This topic is central in the international policy agenda as sustainable communities will better accommodate the challenges of the future putting humans at the core of the process thus enhancing prosperity and welfare.

## 2. Literature Review

### 2.1. Sustainable Innovation Ecosystems

Rapidly obsolescent assets and constant shifts in demand in a globally competitive environment are putting pressure on both the industry and national innovation systems. Trading blocks, nations, regions, and clusters worldwide face ongoing structural changes trying to appraise global innovation trends and technological shifts. Accommodating this hectic pace demands efficient and reinforced innovative ecosystems.

The concept of innovation ecosystem gained increased popularity over the last decade, due to its particular link to open innovation. The term was firstly coined by Tansley [5], to name one ecological element embedding the living creatures and their environment. Moore [6] revived the concept to describe a framework of players in coopetition, highlighting the geographic dimension of knowledge spillover sharing. This increased popularity put the debate on its relevance and definition. Presently, the innovation ecosystem comprises a multilayer framework in which institutions interconnect to develop and share information and knowledge required for the development of new innovation processes [4]. It evidences the co-creation and sharing of firms to provide a coherent solution to meet the challenges of the demand.

Innovation is strongly connected to problem solving, and presently, the challenges relate to complex problems demanding structural changes in individual and collective living such as sustainable development. According to the World Commission on Environment and Development (1987) [7], this state is a dynamic process of change, allowing the present exploitation of resources, the completion of investments, and the path of technological and institutional change, combining the welfare maximization of present and future generations.

Sustainable innovations will work as catalysts for cleaner production, meeting societal challenges in both the short and long run, encompassing economic and environmental targets in local and global dimensions. Sustainable development practices provide background for any context in which humans and the environment are found. These innovations will underpin sustainable development relying upon the networks, local communities, and corporate sustainability as think tanks developing benign solutions to societal challenges [8,9].

As a consequence, sustainability-driven innovation consists of developing goods and services that raise present welfare while efficiently allocating the endowments of resources for both the present and future generations [10,11].

The innovation ecosystem is a network of relationships combining actors and objects that establish connections, both complementary and substitute reinforcing the importance of the institutions and the environment, providing information and knowledge flows through systems of value co-creation,

enhancing sustainability [9]. Sustainable innovation relies on sustainable development, encompassing ethical, social, economic, and environmental principles [4,8].

Sustainable innovation deems sustainable welfare an intrinsic value and income generation an instrumental value. Focusing on both local and global communities and the *innovative milieu* is the ecosystem and not the national system of innovation [8].

The ecosystem will consist of a dynamic, interactive network embedded in an innovation *mindset*, an interactive set-up focused in knowledge creation and diffusion. These ecosystems might be virtual due to the digital transformation we are facing globally; however, they need some grounded hub as members need to physically meet to interact and co-create, to develop new ideas benefiting from their multidisciplinary skills and competences [10].

A vigorous innovation ecosystem will provide firms an innovation environment of "tropical rain forests" where they can share value with a community with shared interests; this process will include governments, the value chain, and the user community, which communicate and promote innovation in order to create valuable new products [11]. It will be reinforced by openness and flexibility, enlarging participation to unusual partners to grasp the knowledge arising from the quintuple helix. Innovative activities are not developed inside the firm borders anymore; they are part of broader interaction with the environment, involving various players embedded within an interdependent innovation ecosystem [12]. These frameworks present a straightforward agenda to bring together human resources boosting entrepreneurial initiatives in a bidirectional way, therefore becoming collective intelligence catalysts. The ecosystem will be revived when fed by external knowledge and contributions, which will spawn an innovative *mindset* [13,14].

Traditional ecosystems tend to centralize in one entity, which benefits the most from the added value; hence, this concentration should be avoided, placing the entire community at the epicenter of the ecosystem. Establishing organized interactions will favor the continuity of the ecosystem, which should be settled on trust, sharing, and a meaningful sense of identity that will consolidate the network based on shared values, which will enhance sustainable practices [12,13].

Sustainability does not come itself; it requires enough resources and capabilities; moreover, present environmental problems call for more environmentally benign technology. The best way to survive market volatility and survive the long run is throughout innovation management and technological innovation to enhance sustainability [15].

Recent theoretical developments such as Reynolds and Uygun [16] argue that inside modern ecosystems there will be high level of interaction between key players such as universities, the value chain, and the user community to create innovative capabilities. This is further reinforced by Song [17], underlining the importance of external ties with suppliers, competitors, and user community within a centralized interaction model.

Shifting from value chains to ecosystems is more prone to organizations adopting industry 4.0 frameworks, service, or customer orientations as they are emerged in networked ecosystems; still, this movement calls for changes in the business model and increasing enrolment with stakeholders [18, 19]. The existence of solid community networks with different roles and interests will generate mutual challenges requiring sustainable practices to uphold the ecosystem.

The consistent emergence of innovations requires a dynamic and sustainable ecosystem encompassing universities and research agencies, financial endowments, sufficient demand, human capital, specialized knowledge, and willingness to collaborate in a global perspective [4,16]. Sustainable innovations add new features to conventional innovations, linked to market-desirable attributes such as durability, locality, resource and energetic efficiency, and reduction of environmental burdens [8]. Performing innovation inside the firm walls is no longer possible given the agile requests of the environment.

Endowments of intellectual capital feed the collective knowledge (explicit or tacit), serving both firms and society to amplify the ability to generate income of other productive factors, reinforcing

competitive advantages. Its existence reinforces the firms' absorptive capacity, embedded in processes and capabilities inside the firm domain, which will also enrich the ecosystem [20].

Innovation tends to cluster among certain sectors or geographies with faster levels of growth and imply structural changes [21]. Indeed, regional development is happening in large clusters, cities, and metropolis. Still, R&D activities, patenting, and value creation occur in globalized innovation hubs. In that sense, smaller regions must complete the effort to identify innovative potential to fully exploit this framework. This process goes along four stages: Inception, implementation, consolidation, and renewal, hence the final stage is not observed in many regions [17].

As a consequence, it is imperative for firms to shift innovation strategy from organization-centered to ecosystem co-creation. This framework will approach organizations and improve sustainable and smart product development based on co-creation, leveraging institutional integrations, and improving the allocation of knowledge and assets inside the ecosystem [22].

Persistent growth in the extraction and use of resources in absolute terms, due to the magnitude of production growth and overconsumption, has led to waste overwhelming resource endowments. Civil society, governance, and private institutions demand for sustainability-oriented innovation systems to increasingly rationalize consumption [23].

Increasing awareness about environmental depletion has pushed innovation towards sustainability in both technological and consumption domains, resulting among others in eco-innovations with positive societal multi-level impacts. The development of these actions relies upon knowledge inflows and outflows exchanged with other agents outside the firm to speed up internal innovation and enlarge the market for innovations with increased value for the environment and society [24]. Innovation has been seen as an important tool for achieving sustainability [25], forming a key binomial in the pursuit of environmental, economic, and social development [26].

Interfirm collaboration is therefore central for sustainability purposes; however, its effect will differ according to both firm and product characteristics such as size, innovation type, elasticity, market share, and production costs [27].

## 2.2. Open Innovation 4.0

Open innovation is an innovation model that relies on the purposeful use of inflows and outflows of knowledge to leverage internal innovation processes reaching new paths to market, as the firms look to advance their technologies. The organization's boundaries become more flexible, permitting the combination of the internal resources with the external co-operators [18]. This model of innovation was firstly proposed in 2003, redirecting the flows of knowledge and the innovation strategies to boost collaboration among firms and other agents inside and outside the value chain, shifting towards a co-innovation paradigm in which the firm speeds up the innovation pace and the organization changes the business model buying and selling knowledge as needed [18,28,29].

Opening the innovation strategy plays a key role towards effective strategic sustainable management. In doing so, firms can leverage knowledge production and management promoting sustainable innovations that retro-feed organizational sustainability. Efforts will be put into knowledge management and the incoming ideas from the external stakeholders, such as research centers, universities, suppliers, and customers. If there is a breakdown of values, in which knowledge arises through partners, the network will acquire relevant skills to manage knowledge and innovation as complements [30].

Blurring the boundaries between the firm and its environment will enable transferring innovations to different marketplaces, with bidirectional knowledge flows circulating outside the organizational borders, highlighting the increased benefit of knowledge sharing throughout partnerships and networks. It implies leveraging external sources of knowledge such as other firms, consumer community, and the ecosystem. In doing so, organizations will combine internal and external know-how, extending the collaboration with the rest of the ecosystem, mostly the Academia and the user community, thus accelerating the innovative process [18,28,29].

Regardless of the centrality of the user community in driving socio-technical transitions, its role within sustainable innovation remains largely overlooked by policymakers. Empirical evidence proves that these agents can no longer be neglected; still, policymakers remain apprehensive about the potential of the user community in this process [31].

Researchers, practitioners, and policy makers quickly understood the importance of the shifting paradigm, and OI was granted important acceptance and diffusion, due to its adherence to reality. Nearly a decade later, the framework was updated arguing that success of the process depends on the knowledge flows and that they should be carefully managed inside and outside the firm boundaries with straightforward mechanisms providing already-established solutions, accordingly to the business model for all kinds of knowledge flows [29].

Despite some skeptical considerations [32], the concept was awarded the trust of the community and remained in solid position, being refined by several authors e.g., [33–36]. Most of the criticisms relied on the need for strong clarification about the agents needing to be involved in the process and their role in the development of the actions [37,38]; however, decentralization in governance is the major challenge put forward by this framework [39]. Innovation is a complex and uncertain process with natural hindering factors; however, open innovation will naturally speed up the pace innovation outputs arise.

Ten years after the concept proposal, Open Innovation 2.0 was reshaped, connecting to the quadruple helix, adding the civil society to the usual players (government, university, and firms), and as a consequence, adding the structural changes driven by user-oriented innovation models; in these frameworks, the speed of the innovation process is accelerated as the different phases co-exist and are set a real world context [3]. The second version of the framework underlines new foundations enhancing the importance of networks and collaborations, promoting interdependencies, relying on corporate entrepreneurship, promoting R&D, and specific intellectual property management, which combined with the accelerated exchange of ideas will boost innovation success, powered by synergies and complementarities [40,41]. The establishment of trusted relations in aligned communities, networks, and stakeholders will be integrated in the surrounding communities thus creating an ecosystem. Innovation 3.0 was proposed in 2010 as conceptual approach, as "*Embedded Innovation*"; the framework encompasses the digital transformation. SMEs (Small and Medium Sized firms) that emerged in a digital and dynamic environment should rely on combined knowledge as it is the most important source of innovation, being essential for survival and growth [42].

This framework captures how companies survive and the way they embed with the other players, focusing on the idiosyncrasies of each. The embedding of the different organisms requires the promotion of the "innovation ecosystems" and business models for innovation to generate sustainable ecosystems. Given the dynamic nature of the innovative process, the organizational process needs to encompass the exploration/exploitation binomial to survive the demanding environment [42,43].

When fed with innovation, embeddedness is a self-sustained process in which the firms along with its stakeholders interact in a certain environment, coexisting and stressing for survival; the process will shape the environment. Mutual influences are exerted, and the innovation process is intertwined with the environment along the innovation life cycle [43].

Embedded innovation practices require the consumer involvement, and the perception of long-term ties between agents creating value and critical development of complementarities among them, seeding the sustainable economies of the future. As a consequence, the framework cannot be considered a substitute but a complement to the conventional models. Structural innovations, despite their importance, are bounded to firms' internal resources, limiting the ability to stay competitive in the long run and beat the competition [42,44].

The improvement of already existing products also generates value to the consumer and the civil society due to the engagement with smart techniques and responsibility. This model will allow for the maintenance of the demanding pace of the innovation locomotive. Additionally, valuing consumer

habits will address the environmental responsibility issues, which are at the core in present policy actions [44]. Embedded innovation therefore builds upon structural innovation.

The creation of sustainable sources of growth will rely on expanding from structural innovation to the consumer-community, tying in with the society. These enlarged communities will encompass diverse people, with different backgrounds, working together towards the creation of an interdependent existence [45]. This paradigm requires potential focus, relationship-based value, and transformational stakeholder engagement to create a sustainable competitive advantage connected to the business model [44]. When moving to the ecosystem level, sustainable innovative practices include the co-creation of knowledge, the engagement of stakeholders, and the value chain to promote improvements in products, technology, and environment [46]. The ideas that feed the innovation process come from people; as a consequence, they need to be stimulated to generate, discuss, and share them. This atmosphere will leverage firm performance and accelerate the innovation cycle, so training people to acquire this *mindset*, creating and recognizing relevant knowledge, will enhance innovation [45]; notwithstanding, larger firms and firms operating in broad marks are more prone to adopt the framework as they have increased awareness about the importance of the environment in their competitiveness [47].The implementation of open innovation in smaller firms is dependent on their persistent managing control over complexity. Due to their versatility, they can be more effective in combining alternative practices, introducing new products and new markets, creating virtuous innovation cycles based upon systematic emergence and partially on complexity control; this will keep the openness of the culture and the business model alive, rising altruism and promoting trust-based collaborations [48].

The integration of the value chain (vertical and horizontal) and the interoperability will break down firm boundaries into a network; this dynamic process will change and create the existing roles of agents. Creating and benefiting from the value emerging from the ecosystem goes beyond the individual value chains, forcing business models to change, shifting to industry 4.0 or alike [49,50]. In this environment, firms will develop new capabilities and meet the consumer-community by means of digital tools gathering and analyzing data to support evidence-based strategies being part of a multi-sided dynamic ecosystem rather than a linear value chain [51].

Industry 4.0 was firstly proposed by the German government in 2011 as a milestone to the fourth industrial revolution. It encompasses multiple advances in digitization, automation, and robotizations. It will transform the value chain to global business model, based on the construction of systems and relationships between machines and machines and humans [1].

This model promotes an accelerated innovation cycle, which accommodates fast-changing consumer expectations, switching to automatization, digitization, and digital security. Consequently, at its core is fostering cooperation and networking among businesses and other entities in the ecosystem to tackle challenges but also to develop new innovations, ideas, or even new businesses [52]. Regardless of the individual characteristics, the open innovation framework will facilitate the creation and understanding of different linkages to be established with players to raise the efficiency of the innovative process, as well as the exchange of resources and knowledge with other entities.

The evident need for transition towards the digital requires a technological push from firms, and with the acceleration of the innovation processes, firms must be able to quickly identify the value of the open innovation processes [53], and leverage their competences to speed up the transition to digital. In the presence of global business models, in which there is constant communication, networking is essential among similar and different players. This exchange of information and knowledge transfer at a macro scale will make the best practices available to everyone, embedding all members with front-edge technology, which will optimize the use of resources, respect the environment, and include community values, being a consequence of sustainable ecosystems.

The literature highlights the existence of an N-shaped curve called the open innovation paradox, which has to be fold back. With the digital transformation, the dynamics of open innovation is increasing rapidly overcoming the inflection point, decreasing the costs of these dynamics, which is in

"rocket-shooting stage" for firms, institutions, and all other players in the ecosystem [54]. The university seems to play a major role in this process, demanding mutual adjustments [55,56]; still, the transition demands for the establishment of alliances [57] and the exploitation of the academic research in industrial innovations [58], as the knowledge produced in universities is increasingly valuable to firms [59].

Developing open innovation strategies intertwines external sources of knowledge with the internal; this dynamic process combines knowledge, human and financial resources, and all other players in the collaborative ecosystem [60]. Specific external supports are therefore required mainly in the case of smaller firms; a good practice in this field is partnership development, multi-dimensional clustering with legal support responding to market changes with prompt sustainable innovations [47].

Open innovation will help turn the immense challenge Europe is facing into an opportunity by investing in the future. European Green Deal the digital transformation initiatives will boost employment throughout innovative and inclusive growth, fostering the resilience of societies environmental sustainability [61].

### 2.3. The Loose Links in OI and the Ecosystem: University-Industry-Collaborations

Knowledge is the masterpiece of the whole open innovation framework, in particular, the external (tacit or explicit), as it relies on the flows; the process requires the coordination of research and development within and integrated horizontal concept to minimize the costs, by means of outsourcing research results developed by the company [30].

Open innovation has been addressed in different perspectives and the conceptual framework has evolved, however the analysis of the patterns of linkage and the associated gains with external collaborations is still overlooked [62], along with the drivers of collaboration and why and with which external entities to collaborate [56,57].

According to Shin et al. [63], six knowledge capacities are required to build an open innovation framework: (1) Transformative capacity; (2) connective capacity; (3) inventive capacity; (4) absorptive capacity; (5) innovative capacity; and (6) desorptive capacity. These competences will allow the retention of internal and external knowledge, its exploration, and exploitation. The connective capacity refers to the ability to establish links with other agents to access external knowledge bases, through inter-organizational relationships such as strategic alliances.

Sustainability is undoubtfully essential these days; nevertheless, going through its path is approached differently among communities. Innovation has recently emerged as a means to achieve sustainability. An integrative capability-based framework (including exploration, retention, and exploitation phases of innovation) based on the classic evolutionary model must be implemented, as sustainability requires more diverse and particular sustainable partners. The effectiveness of the innovation process depends on the context, enabling the combination of open innovation capabilities with the specific context of sustainable development [64].

Universities are at the center of the innovation process given their role in educating students as agents of innovation, promoting and inspiring their critical spirit [65], and transferring their knowledge to promote organizational aptitudes and development tools, which are extremely valuable assets inside the organizations [66,67].

The analysis of the economic performance along with the innovative strategies of developed economies reinforces the importance of knowledge production and diffusion in different technological regimes as a booster of competitive advantages [68]. According to Xie and Wang [14], there are six principal modes by which firms engage in open innovation ecosystems: (a) Firm–university–institute cooperation, (b) interfirm cooperation, (c) firm intermediary cooperation, (d) firm-user cooperation, (e) asset divestiture, and (f) technology transfer; here, the focus will be put on the first.

As a consequence, university firm collaborations play a determinant role in the promotion of those flows, promoting enlarged exchanges in seminal and specific domains [56–58]. The challenges of the future and the empowered role of the Academia in the ecosystem demand institutional and

bureaucratic adjustments. The active enrolment in research projects, which will shift the priorities and the financing incentives of the institutions, will force them to abandon the ivory tower fostering closer relations with the entrepreneurial sphere [55]. University–industry collaborations encompass a multi-layer framework of collaborative research, scientific consulting, or research contracts intertwining the theoretical and real-world dimensions [69,70].

The interconnectedness of universities and firms will backup policy frameworks such as the Research and Innovation Strategy for Smart Specialization (RIS3) harmonizing regional and national innovation ecosystems with the helix [71], with multiple influences among institutions with constant and bidirectional influences sustaining the entrepreneurial ecosystem with the richness of a reef [72]. At present, the dissemination of the digital transformation is eroding the traditional boundaries of the industry, requiring the redesign of the existing Business Models [44].

The Fourth Industrial Revolution is considered as the most powerful innovation driver at present, causing an entire innovative wave; features such as real-time capability, interoperability, and the horizontal and vertical integration of production systems through ICT systems will soon become a reality [52]. Meeting this transformation is urgent for firms to survive a globalized competition and the ever-changing demand, which shortens innovation and product lifecycles demanding for a more interactive environment [50].

The shape and intensity of the collaborations will be strongly influenced by the university research resources along with the competences of the research teams and the institutional orientation for the commercialization of science [67,69,73].

More than ever, universities and firms will collaborate based on the belief that their complementarity will reinforce the strength and the gains concerning they partnership, given that the absence of competition will prevent opportunistic behaviors, reinforcing trust [57,69] and combining different endowments of physical and human resources with expertise [69,74]. Very often, these collaborations pushed universities to move from traditional to entrepreneurial organizations [75,76].

Traditionally, universities connect to knowledge transfer by means of education. Skilling the labor force will bridge the knowledge from the inside to the outside world. Through research, they become engines of knowledge creation and diffusion, producing new technologies and tools useful for the entrepreneurial agents, making fruitful connections. These R&D dynamos bridge scientific knowledge to applied knowledge by means of scientific networks, which will help the firms in speeding up their innovation cycles and decreasing innovation costs [71,72,76]. The knowledge transfer should incorporate the regional needs consequently creating positive *spillovers* to the local community. Nowadays, the university cannot be detached from its ecosystem, as it must commit to the community needs in the generation of sustainable actions and welfare. Moreover, most universities draw upon public funding, which means that they shall give back to tax payers, and firms may use their facilities and technology to reduce their costs, tethering internal R&D with the external knowledge. The last decades have shown an exponential increment of this concern, and presently, the third mission is the touchstone with the innovative ecosystem multiplying the extent of the public policies [59,65,71].

The role of Universities in the promotion of open innovation strategies goes in line with its missions as fostering more efficient innovation cycles including public R&D, in this line mutual enhancement, expectably the innovation ecosystems will raise. The metamorphosis of the university into an entrepreneurial institution is inevitable [51], and the new innovation policy frameworks are now assessed in terms of social coordination rather than market oriented [57,62]. However, public research is unique due to its independence and the distancing with commercial leitmotifs having the ability to radically change the landscape of the industry.

In addition to knowledge generation, universities will train their students, supplying the industry with graduated individuals [55,66]. Hence, the university also benefits from its involvement with the industry gaining awareness of the real-world concerns and technological trends, and finding occupations for their students [57,58]. The role of the universities in regional and national systems of innovation is widely accepted, thus the policy actions and instruments to tackle this issue is fuzzy.

Recently, policy makers have concentrated, in a semi-myopic way, on the university's third mission, neglecting to some extent, the others. These institutions are asked to teach research and contribute to the civil society through knowledge creation and diffusion [59,65].

Open innovation helps in framing the entrepreneurial mission of universities as it promotes networking between the heterogeneous players to transfer and commercialize knowledge [42,46]. Loaded with enlarged missions, universities are seen as the holy grail for the development of open innovative ecosystems [42]; with the increased demand for technological progress and disruptive innovations, firms operating in globalized markets have an increased incentive for companies to exploit collaborations with universities [51], given their value as sources of information for innovation [53].

Policy makers consider universities as knowledge crafters, at the center of the ecosystem catalyzing connections and flows of valuable information that will enhance the regional ability to generate income [20]. Multilateral connections appear, in the sense of university–industry–government collaborations, to promote the skilling of the workforce, knowledge diffusion, and sharing with the involvement of the civil society [56].

The promotion of sustainable innovation ecosystems is a co-creative process in which players must contribute and benefit from knowledge creation relying upon absorptive capacities and improvements [76]. In this symbiotic process, the role of each agent overlaps mutually benefiting from different competences that will complement each other.

The university's third mission is leveraged by facilities such as science parks, industrial clusters knowledge transfer offices, incubators, and other infrastructures, which are placed nearby to simplify the link between the actors [72]. Consequently, shifting from conventional structures to networked structures has been the priority of many institutions [73,76]. At present, research projects are often co-subsidized by public and private institutions, with private funding assisting government-funded projects and vice versa [58].

However, the emerging entrepreneurial universities are highly focused on university–industry collaboration and may find themselves relatively limited to the type of research carried out at the institution. Ongoing research may be subject to the requests of the private agents with a solution-driven logic, which will serve the purposes of the ecosystem in the short term, consequently restricting curiosity-driven experimentation and random research performed by scientists [75].

This transformation possibly endangers basic research and, with no investments in long-term incremental R&D, biased profitable initiatives will emerge, as no incentive schemes are created for non-commercial research. Additionally, the paid interest of industry is the existing research that releases the government from supporting public research, as universities become less dependent on public funding, arguing in favor of university funding closely linked to its economic impact [58,65]. The situation will worsen as the enrolled private agents, either as sponsors or as free-riders, become successful in appropriating the knowledge produced by the public sector, with the contribution of tax payers, patenting the findings and internalizing monopoly profits on the one hand and preventing the general spread of the progress on the other [62,65].

The implementation of generalized open innovation strategies in the ecosystem towards the digital transformation process will approach different players, placing them in convergent technological paths. The reinforcement of this paradigm by means of public policy, with the active intervention of all agents in the ecosystem, is fundamental to achieve long-term sustainability, guaranteeing the continuity of regions.

## 3. Methodology

To measure the role of open innovation strategies in the promotion of sustainable innovative ecosystems, five logit models were run. In doing so, empirical evidence will be gathered to discuss the impact of the open innovation strategy along with reliance on public funds and the user community, as well as firm structural characteristics that impact the propensity to perform the different innovation types, and address the role of the innovation sources, such as the universities, as enhancers of the effect.

### 3.1. Database Description

Empirical analysis will rely on data from the Community Innovation Survey (CIS) 2016, from Portugal, covering the biennia of 2014–2016. The database includes 6775 firms operating in Portugal with heterogeneous structural characteristics and innovative profiles. This survey is the most comprehensive concerning innovation-related issues providing enlarged evidence for the relevant variables in use.

Innovative strategies take time to produce effects, and innovation cycles are long. The efforts performed by this period will hopefully produce the desired effects in the future. Addressing the case of Portugal is of particular interest in this period as it is amongst the regions that have made the greatest evolution in the recent years, being classified in 2019 and 2020 as a "strong innovator". To the European Innovation Scoreboard, transition towards the leading stage was reached only in a group of seven countries. In Portugal, these years were the greatest "leaps" after a persistent classification as "moderate innovator"; additionally, its leading capacity was highlighted in domains such as "innovation in SMEs" [61].

### 3.2. Exploratory Analysis

To provide a detailed description of the variables in use, Table 1 addresses a description of each variable and its scale of measurement. In most cases, the measurement follows the CIS original scale; in others, simple mathematical conversions were performed.

**Table 1.** Variable description.

| VARIABLES | DESCRIPTION | MEASUREMENT |
|---|---|---|
| TEC_REG (1) | Technological regime of the firm (according to Boliacino and Pianta) [77] | 1 = supplier dominated; 2 = scale intensive; 3 = specialized supplier; 4 = science based |
| SIZE (2) | Firm dimension | 1–3 degree |
| EXP_PROP (3) | Proportion of the turnover exported | Decimal |
| EMPUD (4) | Human Capital intensity | 1–6 degree |
| OPEN (5) | Performing inbound and outbound innovation | Binary |
| FUNDS (6) | Beneficiary of funds | 1 to 4 count |
| SUNI (7) | Relying upon Universities as source of innovation | Binary |
| USER_COMM (8) | Relying upon User Community as source of innovation | Binary |
| HSI_LEG (9) | Concerned about Hygiene and Security legislation | Binary |
| ENV_LEG (10) | Concerned about Environmental legislation | Binary |
| INT_ASSET (11) | Having registered copyrights or others | Binary |
| PROD_INNOV (12) | Having performed product innovation | Binary |
| PROC_INNOV (13) | Having performed process innovation | Binary |
| ORG_INNOV (14) | Having performed organizational innovation | Binary |
| MKT_INNOV (15) | Having performed marketing innovation | Binary |
| TECH_INNOV | Having performed product or process innovation | Binary |

Concerning technological regimes, firms were divided into four categories, according to technological intensities. Firm dimension encompasses small, medium, and large, following the European Commission methodology and the European Innovation Scoreboard [61]. Human capital was divided into intensities, following the CIS methodology and the EIS [61]. Innovation types were appraised as binary variables, depending on the firm response. The proxies in use follow the survey methodology and are further described in Table 1, and were similarly used in [78].

Table 2 provides the descriptive statistics and the correlations among the variables in use. Regarding correlation, moderate intensity is found, which highlights the accuracy of the set used and guarantees the inexistence of multicollinearity. This was further reinforced with VIF (variance inflation factor) tests, which pointed in the same direction.

**Table 2.** Descriptive statistics.

| VARIABLES | Mean | S.D. | Min | Max | (1) | (2) | (3) | (4) | (5) | (6) | (7) | (8) | (9) | (10) | (11) | (12) | (13) | (14) | (15) |
|---|---|---|---|---|---|---|---|---|---|---|---|---|---|---|---|---|---|---|---|
| TEC_REG (1) | 1.753 | 1.100 | 1 | 4 | 1.00 | | | | | | | | | | | | | | |
| SIZE (2) | 1.351 | 0.572 | 1 | 3 | 0.05 | 1.00 | | | | | | | | | | | | | |
| EXP_PROP (3) | 0.217 | 0.319 | 0 | 1 | 0.06 | 0.24 | 1.00 | | | | | | | | | | | | |
| EMPUD (4) | 2.413 | 1.830 | 0 | 6 | 0.44 | 0.15 | 0.04 | 1.00 | | | | | | | | | | | |
| OPEN (5) | 0.103 | 0.304 | 0 | 1 | 0.11 | 0.22 | 0.14 | 0.20 | 1.00 | | | | | | | | | | |
| FUNDS (6) | 0.295 | 0.700 | 0 | 4 | 0.13 | 0.18 | 0.19 | 0.17 | 0.34 | 1.00 | | | | | | | | | |
| SUNI (7) | 0.476 | 0.499 | 0 | 1 | 0.14 | 0.20 | 0.09 | 0.23 | 0.23 | 0.21 | 1.00 | | | | | | | | |
| USER_COMM (8) | 2.363 | 2.383 | 0 | 6 | 0.10 | 0.13 | 0.11 | 0.21 | 0.23 | 0.27 | 0.41 | 1.00 | | | | | | | |
| HSI_LEG (9) | 0.718 | 0.483 | 0 | 5 | 0.08 | 0.12 | 0.10 | 0.20 | 0.21 | 0.26 | 0.03 | 0.62 | 1.00 | | | | | | |
| ENV_LEG (10) | 0.722 | 0.489 | 0 | 5 | 0.07 | 0.12 | 0.10 | 0.19 | 0.22 | 0.27 | 0.06 | 0.61 | 0.96 | 1.00 | | | | | |
| INT_ASSET (11) | 0.289 | 0.705 | 0 | 6 | 0.13 | 0.11 | 0.15 | 0.20 | 0.26 | 0.28 | 0.18 | 0.24 | 0.20 | 0.19 | 1.00 | | | | |
| PROD_INNOV (12) | 0.335 | 0.472 | 0 | 1 | 0.04 | 0.10 | 0.17 | 0.08 | 0.26 | 0.29 | 0.08 | 0.40 | 0.45 | 0.45 | 0.23 | 1.00 | | | |
| PROC_INNOV (13) | 0.495 | 0.500 | 0 | 1 | 0.03 | 0.14 | 0.12 | 0.10 | 0.25 | 0.29 | 0.03 | 0.48 | 0.62 | 0.61 | 0.16 | 0.39 | 1.00 | | |
| ORG_INNOV (14) | 0.358 | 0.480 | 0 | 1 | 0.10 | 0.12 | 0.11 | 0.22 | 0.24 | 0.23 | 0.15 | 0.43 | 0.47 | 0.47 | 0.23 | 0.27 | 0.40 | 1.00 | |
| MKT_INNOV (15) | 0.396 | 0.489 | 0 | 1 | 0.05 | 0.04 | 0.01 | 0.20 | 0.20 | 0.21 | 0.12 | 0.49 | 0.50 | 0.50 | 0.26 | 0.35 | 0.37 | 0.42 | 1.00 |

## 4. Econometric Analysis

### 4.1. Econometric Estimations

In order to appraise the determinants of the innovative performance for each innovation type, five logit models were run, which are presented in the following Table 3. Models 1 and 2 refer to the conventional types of innovation: Product and process, respectively. Model 3 encompasses having performed any of the types of technological innovation (product or process). Non-technological types of innovation such as marketing or organizational appear in Model 4 and Model 5, respectively. Open innovation studies focus on product and process innovations; however, given the emergent importance of the other types, analysis was further extended.

**Table 3.** Econometric estimations—marginal effects after logit estimation.

| VARIABLES | PROD_INNOV | PROC_INNOV | TECH_INNOV | ORG_INNOV | MKT_INNOV |
|---|---|---|---|---|---|
| TEC_REG | −0.053 * | −0.109 *** | −0.184 *** | −0.035 | −0.137 *** |
| | (0.032) | (0.037) | (0.053) | (0.032) | (0.033) |
| SIZE | −0.110 * | 0.216 *** | 0.051 | 0.061 | −0.218 *** |
| | (0.057) | (0.072) | (0.103) | (0.058) | (0.059) |
| EXP_PROP | 0.726 *** | 0.242 * | 0.592 *** | 0.182 * | −0.520 *** |
| | (0.105) | (0.129) | (0.205) | (0.104) | (0.107) |
| EMPUD | −0.082 *** | −0.101 *** | −0.181 *** | 0.145 *** | 0.144 *** |
| | (0.021) | (0.024) | (0.035) | (0.021) | (0.021) |
| *OPEN* | *0.536 *** | *0.430 *** | *0.422 ** | *0.554 *** | *0.375 *** |
| | (0.098) | (0.125) | (0.190) | (0.098) | (0.101) |
| FUNDS | 0.188 *** | 0.181 *** | 0.180 ** | 0.106 ** | 0.091 ** |
| | (0.044) | (0.055) | (0.085) | (0.043) | (0.045) |
| *SUNI* | *−0.028* | *−0.049* | *0.101* | *0.004* | *−0.141 ** |
| | (0.074) | (0.089) | (0.130) | (0.074) | (0.076) |
| *USER_COMM* | *0.114 *** | *0.094 *** | *0.080 *** | *0.161 *** | *0.254 *** |
| | (0.016) | (0.020) | (0.029) | (0.017) | (0.017) |
| HSI_LEG | −0.050 | 0.560 | 0.665 | 0.072 | 0.003 |
| | (0.241) | (0.349) | (0.501) | (0.243) | (0.247) |
| ENV_LEG | 0.365 | −0.282 | −0.640 * | 0.091 | 0.103 |
| | (0.229) | (0.267) | (0.345) | (0.219) | (0.227) |
| INT_ASSET | 0.346 *** | −0.033 | 0.134 | 0.300 *** | 0.527 *** |
| | (0.049) | (0.052) | (0.088) | (0.047) | (0.054) |
| Constant | −0.551 ** | 0.780 ** | 2.606 *** | −1.432 *** | −0.685 *** |
| | (0.243) | (0.332) | (0.455) | (0.239) | (0.242) |
| Observations | 4229 | 4229 | 4229 | 4229 | 4229 |

Standard errors in parentheses: *** $p < 0.01$, ** $p < 0.05$, * $p < 0.1$.

Given the binary nature of the dependent variable (whether or not having performed innovation), a binary count model (logit) was implemented. Notwithstanding, most of the empirical evidence

presents different frameworks in terms of the dependent variables or establishes international comparisons, which precludes direct comparisons.

Logit estimations were omitted due to the impossibility of interpreting the coefficients; Table 3 presents the marginal effects that quantify the impact on the propensity to innovate caused by changes in the independent variables.

Among the five models, the dependent variable does change to address the impacts of the exogenous variables in each type of innovation. Explanatory variables and controls are the same among the five models in use, allowing for inter-model comparisons.

## 4.2. Results and Discussion

Firms belonging to more knowledge intensive technological regimes are naturally more prone to develop innovative activities irrespective of the innovation type [77,79]. Empirical evidence proves to be contrary in this case, with firms in more knowledge intensive segmentations being less prone to innovate, and the impact being higher in technological innovations. This result is somehow deceptive as the policy makers tend to favor these areas.

There is a positive correlation between firm size and innovation; larger organizations look for appropriating and binding all the relevant ideas that lie beyond their boundaries [80,81]. Contrarily from expected, size increments reduce the propensity to perform product or marketing innovation; conversely, concerning process innovation (Model 2), size raises the innovative propensity. These results evidence that policy makers cannot generalize policy requests based upon size.

Export propensity plays a positive effect on the propensity to innovate regardless the innovation type, with the exception of marketing innovation. Firms operating in external markets are more prone to have a dynamic innovative strategy; as a consequence, incentives promoting the access to foreign markers will reinforce innovative behaviors.

Concerning human capital intensity, the effect on technological innovations (product and process innovations) contradicts previous expectations [45], as enlarged stocks of employees with an undergraduate or more will deter innovations. On the contrary, non-technological innovations such as organizational or marketing will be enhanced by human capital intensity. Results evidence the fact that technological innovations are complementary enhanced physical and human capital.

Public funding raises the probability to perform innovation, regardless of the innovation type [58]; this result reinforces the importance of the conventional policy instruments in the promotion of innovation cycles.

Pursuing open innovation strategies, and their role promoting innovative ecosystems supporting the digital transformation [53], are the core of the empirical analysis. These strategies were proxied by the combination of inbound and outbound R&D flows; and appear with a strong positive impact in innovative propensity for all innovation types. In detail, firms performing open innovation strategies are 42.2 percentage points more prone to develop technological innovations than their non-innovative counterparts. Even for non-technological innovation, being open is still a strong booster raising the probability to innovate by 55.4 pp and 37.5 pp in organization and marketing domains.

Relying upon the university as a source of information for innovation is, according to the literature, an enhancer of the innovative performance [9,70], however the variable fails to be statistically significant notwithstanding the innovation type. The lack of significance of this variable needs to be further appraised; however, these results go along with previous research for the Portuguese case [78]. Despite the positive evolution of the innovation performance as a whole [61], more needs to be done to promote University industry collaborations.

The effect of the augmented helix in innovative performance demands user-community involvement and the civil society along with the adoption of responsible practices in innovative strategies [19]. The influence of the user-community on the innovation initiatives was measured by a binary variable, which takes the value 1 if the firm indicates having considered the user opinion and further contributions to develop innovations. In all dimensions of analysis, the variable appears

with a positive impact in innovation. This result points towards the existence of co-creation dynamics encompassing the knowledge flows emerging from the ecosystem, retro-feeding innovation practices.

Social responsibility dimensions were appraised by means of legal aspects such as the hygiene and security along with the environment [73]. In general, these vectors are not yet relevant as innovation determinants. These results should get the attention of policy makers reinforcing the urgency for the redesign of these regulations. Sustainability strongly relies on public policy and its accuracy in generating desirable behaviors; as a consequence, the promotion of corporate responsibility cannot be neglected.

## 5. Concluding Remarks

### 5.1. Theoretical and Empirical Implications

Innovation is the main driving force for sustainable development and the promotion of growth; as a consequence, the topic gained centrality in the agenda of policymakers, practitioners, and researchers. So, innovation and sustainability are considered, to some extent, two dimensions of the same reality [82,83]. Regarding international regulations in the field of environmental protection, the focus was put on the preservation of natural resources [10].

Drawing on a cross-sectional sample of firms operating indifferent sectors and running several logistic models to identify the determinants of the different innovation types, empirical evidence proves that open innovation, operation out of boundaries, public funding, and the user community promote innovation cycles. Moreover, technological regimes and dimension produce mismatched effects. This implies that economic and sustainability innovation goals can be attained at once.

As industrial innovations are becoming increasingly more open, it is important to understand where open innovation will add value to knowledge intensive processes. The digital transformation demands for connections, networks, and high speed in the innovation cycles. The present paper contributes to the identification of the importance of this new player recently identified and overlooked in the extant literature [37,83–85].

As we journey, further efforts need to be made to systematically include the user community in the ecosystem, given its increasing importance [86]. The empirical findings underline that having the awareness and the proximity to these agents will avoid mistakes being made, as they are the entry gate to marketplace acceptance.

Additionally, the article theoretically and empirically identifies open innovation strategies as innovation boosters regardless of the innovation type, therefore grounding innovation ecosystems. Given the importance of adopting permeable organizational boundaries, managers should focus on this strategy, transforming their business models to meet the fast-changing requirements of the consumer-user community, combining internal endowments with the external, accelerating the innovation cycle, and reducing its costs.

Universities play a key role in creating human capital and in generating new knowledge through research. The article addresses the three missions of the university, and, contrarily to the previous, in empirical terms, the first mission seems to be properly established, as the human capital intensity influences innovative propensity.

Conversely, the connection with the university as a source of relevant knowledge to innovation fails to affect the innovation strategies. This unsettling fact deserves further attention and the identification of the adjustments to put universities at the center of knowledge networks. Nowadays, universities play a determinant role in the advance of basic research while in the 1980s, that mission was led by firms' R&D departments. Business will drain all possible appliances for their knowledge. Still, the firm focuses on their own business model whilst society seizes competition between different ideas. So, new R&D strategies are emerging and need further analysis.

Digital transformation will multiply interactions and sources in unprecedented ways, and open innovation will provide the opportunity to ground these connections, empowering users, optimizing

the match between demand and supply, and minimizing waste. Competition pressure will come from the value chain and the empowered user [86], and this shift demands accurate open innovation strategies. In accordance to Chesbrough [82], the empirical evidence proves the need for addressing the intellectual property issues under the open innovation *mindset*, to meet the singularities of the innovation types and the changes brought by the cooperative innovation processes.

### 5.2. Limitations and Future Research

The importance of the value chain along with the Academia in the promotion of the innovation ecosystems has grasped the attention of academics, practitioners, and policy makers since the 1980s with the introduction of Etzkowitz's helix. As society evolves along with the technology, new players should be considered. The present article shed some light on the role of the user community, however the next generation of studies in the field needs to further explore the role of the digital revolution in the reshaping of the knowledge creation and diffusion processes and mostly the emergence of a central player: The user community.

Nowadays, consumers claim for sustainability requirements; traditional price competitive models are insufficient and the active involvement in environmental protection and social responsibility is demanded by stakeholders. In this vein, sustainability-oriented innovations are the new core of entrepreneurial innovative strategies [10,82]. The present research does not measure the impacts on the firm performance, which deserves further attention.

Despite the numerous studies on university–industry collaborations, there is still insufficient evidence, concerning the upstream, midstream, and downstream aspects of this connection. It is also important to empirically address the role of open innovation as facilitator of this process. This loose link is probably deterring the development of more efficient innovation ecosystems [20,87,88] in the Portuguese case [78]. The question that remains unanswered is: Why does the university fail to impact innovative propensity? This finding should be further analyzed to shed some light in what is hampering the establishment of solid connections between the Academia and the Industry. Open innovation is undermined if this pillar keeps lacking. All in all, practitioners and managers should be aware of the importance of this source of knowledge for their processes.

A more detailed analysis is required to understand the positive effect on non-technological innovation compared to the unexpected negative effect on technological innovations. The variable in use is perhaps broad, encompassing general degrees rather than specific competences being a limitation. Apparently, skilling is biased towards non-engineering competences. Separating the number of engineers from the grand total is impossible in the CIS database, but it could provide a finer conclusion. Despite the robustness of the respondent sample, present results emerge from a sectional analysis and they may represent an exceptional coordination rather than a long-term trend. Running the same empirical analysis in a diachronic perspective would reinforce the findings, which is an open avenue of research for future works.

Regardless of their structural characteristics, firms increasingly rely upon external sources of knowledge rather than closing themselves off by being confined to the exploitation of internal resources. However, perhaps due to uncertainty and complexity of the legal framework, small and medium-sized firms are less confident in open innovation networks, anticipating appropriability problems more often than their larger counterparts. As these organizations are the backbone of the industrial fabric in most countries, policy makers should address their weaknesses with a more effective legal system protecting property rights. Policy action should help SMEs in finding more modern business models, promoting the digital transition. Sharing best practices in open innovation networks underlying the extant gains will certainly encourage the adhesion to the ecosystem.

### 5.3. Policy Recommendations

In a globalized era with proliferating open innovation practices, policy makers must ensure a democratic access to knowledge and technology. Policy changes will ascertain firms and entrepreneurs

towards co-creation. Open innovation is the key for supporting networks rather than individual firms in the promotion of market competition.

The new innovation policy must abandon the centrality of large companies and consider the roles of human capital, competition, financing, intellectual property, and public data in promoting an open innovation ecosystem encompassing smaller organizations.

In most cases, current policy instruments still rely upon the closed innovation paradigm, focusing upon large markets, traditional sectors; protect national companies; and subsidize the larger organizations.

Empirical findings sustain that large firms are potentially starting to outsource innovation and entrepreneurship projects in environments that are considered more open and more agile, as their size contributes to their lower propensity to innovate. It is worth mentioning that the Portuguese achievements in terms of SME dynamism towards innovation were highlighted in the European Innovation Scoreboard 2020 [65]. Again, policy makers' attention is deserved as often large firms are positively discriminated towards public funding.

The insignificant results of the environmental and hygiene legislation in and security domains should be further analyzed, as they are strongly tied to sustainable practices and responsible attitudes towards resource use. Designing demanding policy measures will discipline industrial practices and standardize desirable behaviors.

The positive effect of funding demand more sophisticated actions such as policy mixes reallocating spending, as technology intensive sectors are not the leading innovative group. Small and medium-sized firms deserve specific supports given their high propensity to innovate. Incentives and subsidization should rely upon technology maturity. Start-ups encompassing immature technologies should be subsidized, and more mature organizations should perhaps benefit from lighter incentives such as fiscal benefits, tax credits, or loan guarantees.

The transition from closed to open innovation requires new funding frameworks combining the strengths and the weaknesses of the ecosystem throughout smarter policy packages such as mandatory consortia to reach public grants. Managers have certainly understood the importance of the public support so they will link to the University.

Natural environment preservation has gained momentum and the international community presented concepts such as "green economy, green growth, and green development". Open innovation effectively deals with market failure correction such as externalities minimizing economic distortions related to economic value and green value. Implementing green governance will foster resource preservation and expectably provide next generations a promising future. Still, this new governance paradigm should avoid the "governance failure" caused by "collective action dilemma" [89].

Developing sustainability-oriented open innovation framework requires a broader stakeholder approach [90]. Governments must promote innovative strategies building upon latent capabilities and focusing on regionally relevant problems, promoting societal mind-changing road maps. Major weaknesses will be surpassed, and the innovation requirements will be identified. Vertically integrated labs are shifting to disintegrated networks of innovation tying agents into ecosystems centralized in the private sector, public action needs to adapt.

Facing the present challenges of the uprising economic crisis with unprecedented consequences, given the simultaneous cut in both aggregate demand and supply, reinforces the need of a knowledge-based model of development [88]. Enhancing regional capabilities will increase cohesion and create self-sustained ecosystems, which will pace up the speed of recovery through inclusive growth [20]. Concerning scientific decision-making processes and the long-term equilibria between man and nature, governance should address sustainability and the concept of green governance should be implemented in a timely manner [89].

Nurturing open innovation ecosystems is vital for the acceleration of recovery, boosting sustainable and responsible practices along with the respect for local communities. This innovation strategy

contrasts with conventional models as, for the first time, it places the human at the center of disruptive innovation with different roles being played.

**Author Contributions:** Conceptualization, J.C.; methodology and empirical analysis, J.C.; formal analysis, J.C.O.M. and J.C.; writing—original draft preparation J.C. and J.C.O.M., writing—review J.C. and J.C.O.M. All authors have read and agreed to the published version of the manuscript.

**Funding:** This research received no external funding.

**Conflicts of Interest:** The authors declare no conflict of interest.

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
