# Peer review of "Open Innovation 4.0 as an Enhancer of Sustainable Innovation Ecosystems"

_sustainability, doi:10.3390/su12198112_

Round 1
Reviewer 1 Report
This is an interesting paper, which needs further revision.
First, the literature review is weak. Open Innovation, Open Innovation 4.0 and Sustainable Innovation Ecosystems must be discussed and defined first. Then, more research has to be integrated, as recently a lot was published in this area. For this, the authors should do a better analysis of related papers linking innovation and sustainability in journals such as Sustainability, Journal of Cleaner Production, etc.
The discussion is also far too short and general. Please provide concrete suggestions for further research, and for managers. Also limitations and future research suggestion should be outlined in detail.
Reviewer 2 Report
Empirical studies about open innovation are one of the most difficult challenges that can be addressed nowadays.
This article is a valuable attempt to develop such kind of studies in the case of Portugal but, as authors admit, “…a more detailed analysis is required to understand the positive effect on non-
technological innovation compared to the unexpected negative effect on technological innovations.”
In order to assess the scientific soundness of your analysis I would like to ask to questions to be clarified by authors.
First, how are related you data of CIS 2016 with the data about 2019 and 2020, if exists (page 8, “Addressing the case of Portugal is of particular interest as it is amongst the regions which have
made the greatest evolution in the recent years, being classified in 2019 and 2020 as a “strong
innovator”, part of a restrictive group of seven countries.” It is unclear on your table.
Second, a better explanation of each relevant variable and its theoretical connections with the other is needed at 3.1 Database description (p. 8).
As a general comment, I would say that your literature review is too large, broad and general. It is very good, but for a research article you should focus on the relevant concepts. For instance, your explanation about the three missions of Universities is being written as a very introductory manual for students. I would suggest narrowing the focus of your review to more technical issues related to your research such a previous findings about methodologies to assess open innovation indicators and statistics.
Round 2
Reviewer 1 Report
I think the revisions was only very minor. Actually I am somewhat disappointed that my recommendations are not integrated to a large extant. Especially the literature review is still very weak missing most of the recent publications from recent years on open innovation. This needs a major revision. Also the implications part is short and general.
Author Response
Many thanks for the helpful suggestions and taking the time to read my article.
All the comments provided were included in the new version of the paper. As suggested, the literature review was improved to accommodate the points of the revision.
Recent articles from "Journal of Cleaner Production" and "sustainability" that were missing in the former version are now part of the article. Please see attachment with track changes.
Hope that this version provided a cleared picture on the topic and meets the expectations presented in your comments.
Thanks again,
Regards
Joana Costa

Round 3
Reviewer 1 Report
Thanks for the chance to look at this revision. The article clearly improved a lot, so thank you to the author(s). However, the literature coverage is still rather weak. Hence, I suggest that recently published papers are included, for instance, if you check on scopus. My main comment is still, that implications for theory and practice must be outlined in detail. So the current concluding remarks should be changed into a discussion to highlight the contribution. Also, limitations and future research must be discussed.
Author Response
Many thanks for the helpful comments provided. They allowed the document to improve.
Following the recommendations:
(1) concerning the suggestion to include recently published papers - additional references were included mainly in the improved sections
(2) implications for theory and practice must be outlined in detail. So the current concluding remarks should be changed into a discussion to highlight the contribution - the entire section was re-written and re-structured as seen in the attached document with track changes.
(3) limitations and future research must be discussed - an entire sub-section was developed to accomodate this request.
Additionally - the results section was also improved to provide a more solid connection to the following.
Hope that the changes are accordingly to expected.
